# Spatiotemporal variation of the maximum cooling effect across edge-to-interior gradients in forest patches of southwestern China

Zhangjian Xie[1], Bin Wang[1]*, Qifei Chen[1], Wenjun Liu[1], Hong Wang[1], Yuxin Ma[1], Weihong Liu[1], Yajie Jiang[1], Wen Liu[1], Yufeng Ma[1], Cameron Proctor[2], Hans J. De Boeck[3], Zhiming Zhang[1]

1 State Key Laboratory for Vegetation Structure, Function and Construction (VegLab) and School of Ecology and Environmental Science, Yunnan University, Kunming, Yunnan, China, 2 School of the Environment, University of Windsor, Windsor, Ontario, Canada, 3 Research group PLECO (Plants and Ecosystems), Department of Biology, Universiteit Antwerpen, Wilrijk, Belgium

* wb931022@hotmail.com

## Abstract

Forest canopies create buffered understory microclimates that differ markedly from adjacent open-area reference conditions. However, how this buffering effect varies near the forest edge, especially between different forest types and seasons, remains poorly understood. In particular, the spatiotemporal dynamics of maximum cooling and the spatial extent of edge influence across forest types remain under-quantified. This study quantified monthly maximum cooling intensity (MCI) and distance of edge influence (DEI) using transect-based in situ air-temperature monitoring at three natural forest sites in southwestern China that span a temperate coniferous forest (CF), a subtropical evergreen broadleaved forest (SF), and a tropical forest (TF). Air temperature was measured at 1.5 m height from the forest edge to 99.5 m into the interior and was paired with a continuously recorded open reference. DEI varied strongly through the year and among sites, ranging from 18 to 77 m in CF, from 13 to 65 m in SF, and from 17 to 59 m in TF. DEI reached its early-summer minimum in May in CF and in June in both SF and TF, while annual DEI was 82 m in CF, 72 m in SF, and 56 m in TF. Within the corresponding interior zones, extreme cooling was strongest in TF (MCI = −8.2°C) and weakest in CF (MCI = −5.8°C). These site-level patterns indicate that both the intensity and spatial reach of extreme cooling are seasonally dynamic along a climatic gradient, which supports edge-aware microclimate mapping, evaluation of interior habitat connectivity, and conservation planning in fragmented forests under climate change.

**Data availability statement:** The Original data are available from the "Forest edge to interior temperature migration data" Zenodo database (https://doi.org/10.5281/zenodo.15855292).

**Funding:** U25A20641, Joint Fund for Regional Innovation and Development of NSFC (https://www.nsfc.gov.cn/ to Z.Z.M.). 32260291, National Natural Science Foundation of China (https://www.nsfc.gov.cn/ to Z.Z.M.). 202205AM070005, Project for Talent and Platform of Science and Technology in Yunnan Province Science and Technology Department (https://kjt.yn.gov.cn/ to Z.Z.M.). 202101BC070002, Major Program for Basic Research Project of Yunnan Province (https://kjt.yn.gov.cn/ to Z.Z.M.). 202303AC100009, Key Research and Development Program of Yunnan Province (https://kjt.yn.gov.cn/ to Z.Z.M.). 2025Y0076, Scientific Research Fund Project of Yunnan Education Department (https://jyt.yn.gov.cn/ to M.Y.X.). KC-24248574, Scientific Research and Innovation Project of Postgraduate Students in the Academic Degree of Yunnan University (http://www.grs.ynu.edu.cn/ to M.Y.X.).

**Competing interests:** The authors have declared that no competing interests exist.

## 1. Introduction

Forest fragmentation and land-use change are increasingly exposing forest interiors to edge effects, weakening forest capacity to buffer against global climate change [1–3]. Globally, an estimated 20% of the remaining forest area lies within 100 m of a forest edge, where microclimatic conditions are strongly influenced by the surrounding environment [4,5]. Microclimates more closely reflect the thermal conditions actually experienced by organisms than coarse macroclimate data, particularly for near-ground taxa such as seedlings and understory herbs [6–8]. Therefore, characterizing the spatiotemporal patterns of microclimate in edge zones has important ecological implications for biodiversity, ecosystem functioning, and species persistence under climate change [9–11].

Forest canopies regulate understory temperature through shading, evapotranspiration, and wind attenuation [12–14]. As a result, temperatures beneath the canopy are typically more stable than in adjacent open areas, a pattern referred to as microclimate buffering [13]. Forest-edge temperature dynamics are often quantified using two complementary metrics, temperature offset ($T_{offset}$) and the Distance of Edge Influence (DEI) [15]. $T_{offset}$ captures the magnitude of cooling or buffering within forests relative to external conditions [15,16]. DEI describes how far open-area influence extends from the edge into the forest interior [2,15,16]. Together, these metrics have clear ecological and management relevance because they delineate thermal buffer zones and provide an empirical basis for estimating the minimum patch size needed to sustain interior microclimatic conditions.

However, most studies emphasize annual or seasonal mean offsets. This coarse temporal perspective can obscure short-lived extremes and therefore limits inference about how forests mitigate heat stress for organisms [17–19]. Growing evidence indicates that brief thermal extremes can be more consequential than long-term means because they directly constrain survival, growth, and regeneration, thereby shaping realized thermal niches and local community composition [20–23]. Yet edge-related extreme temperature buffering remains insufficiently characterized across forest types and temporal scales, both in its intensity and in its spatial extent. In particular, DEI is commonly estimated at a single temporal scale even though edge-to-interior coupling can vary substantially within a year. DEI derived from monthly, seasonal, and annual data may therefore differ markedly within the same forest, and this scale dependence has rarely been evaluated.

In addition, reported DEI values remain difficult to synthesize across regions. Typical ranges have been suggested for selected biomes, including approximately 20 m in tropical forests and at least 12.5 m in temperate forests [2,16]. However, the empirical evidence is scattered and methodological choices differ among studies, which limits cross-study comparability and leaves systematic assessments across forest types and climatic zones scarce. This gap is particularly evident for subtropical evergreen broadleaved forests. These forests account for more than 11% of global forest cover and are concentrated in Asia where macroclimatic seasonality is pronounced, yet edge microclimate processes remain relatively understudied [11,24].

Southwestern China offers an exceptional setting to address these gaps. Over a relatively compact geographic area, the region spans tropical, subtropical, and temperate forests, creating a strong climatic gradient that allows forest-type effects to be examined alongside variation in background climate [25,26]. The region is also highly fragmented, with more than 32% of forest area located within 100 m of a non-forest edge, reflecting complex topography and land-use legacies [27]. Such pervasive edge exposure can reshape within-forest thermal regimes and may increase ecological vulnerability [2,16,28,29]. Because temperature strongly governs physiological performance, demographic rates, and species distributions [30,31], it is important to understand how macroclimate and edge-related forest structure jointly regulate both the strength of forest cooling and how far it extends into the interior. Quantifying the intensity and spatial reach of maximum cooling along forest edges can support management decisions that require spatially explicit guidance, including buffer-width design, climate-smart retention patches, and the identification of microrefugia in fragmented landscapes [32–35].

In this study, we combine transect-based in situ monitoring with high-frequency temperature records to examine how extreme cooling and its spatial reach change from the forest edge to the interior across three representative forest types in southwestern China, including temperate coniferous, subtropical evergreen broadleaved, and tropical forests. We focus on maximum cooling intensity (MCI) and the Distance of Edge Influence (DEI), and we explicitly evaluate whether inferences about DEI depend on the temporal scale at which it is estimated, ranging from monthly to seasonal and annual aggregations. We address three aims:

1) we quantify the edge-to-interior gradient in MCI and test whether its strength and spatial pattern differ among forest types and across temporal scales;

2) we characterize short-term variation in both MCI and DEI and identify the timing of monthly peaks and troughs that indicate shifts in forest–open-area coupling through the year;

3) we delineate interior zones using DEI-based thresholds and test whether these zones maintain thermal regimes that are statistically distinct from edge conditions, which provides evidence for climatically decoupled microrefugia.

## 2. Materials and methods

### 2.1. Study area

The study area spans the north-south axis and encompasses various climatic zones within Yunnan Province (Fig 1a). The tropical forest (TF) is situated at the Botanical Garden in Menglun Town, Mengla County, Xishuangbanna Dai Autonomous Prefecture (21° 54′ 37.84″ N 101° 16′ 24.59″ E), characterized by a mean annual precipitation of 1493 mm, a mean annual air temperature of 21°C, and an elevation ranging from 470 to 2430 meters. The forest is bordered by grasslands, with the dominant tree species including *Phoebe lanceolate* (Wall. ex Nees) Nees and *Pittosporopsis kerrii* Craib, among others. The subtropical evergreen broad-leaved forest (SF) is located in the Rhododendron Dam of Xujia Ba, Jingdong County (24° 32′ 10.42″ N 101° 0′ 34.78″ E), with a mean annual precipitation of 1882 mm, a mean annual temperature of 11°C, and an average elevation of 2410 meters. The forest edge is contiguous with grasslands, and is primarily dominated by species such as *Eriobotrya bengalensis* (Roxb.) Hook. f., *Ilex szechwanensis* Loes. and *Ilex corallina* Franch. Temperate coniferous forests (CF) are found in the Pudacuo National Park near Shangri-La (27° 50′ 41.89″ N 99° 59′ 8.14″ E), characterized by a mean annual precipitation of 817 mm, a mean annual temperature of 4.2°C, and an average annual elevation of 3827 meters. The forest is also bordered by grassland, primarily dominated by *Abies georgei*. All sites are located within protected areas or long-term research plots and represent long-unharvested stands. No stand-replacing disturbances have been recorded in recent decades, including wildfire, severe windthrow, or major insect and pathogen outbreaks. This operational definition excludes stand-replacing events but does not imply the absence of natural gap dynamics. Our inference is therefore limited to forests with these characteristics, which are well represented within the regional protected-area network.

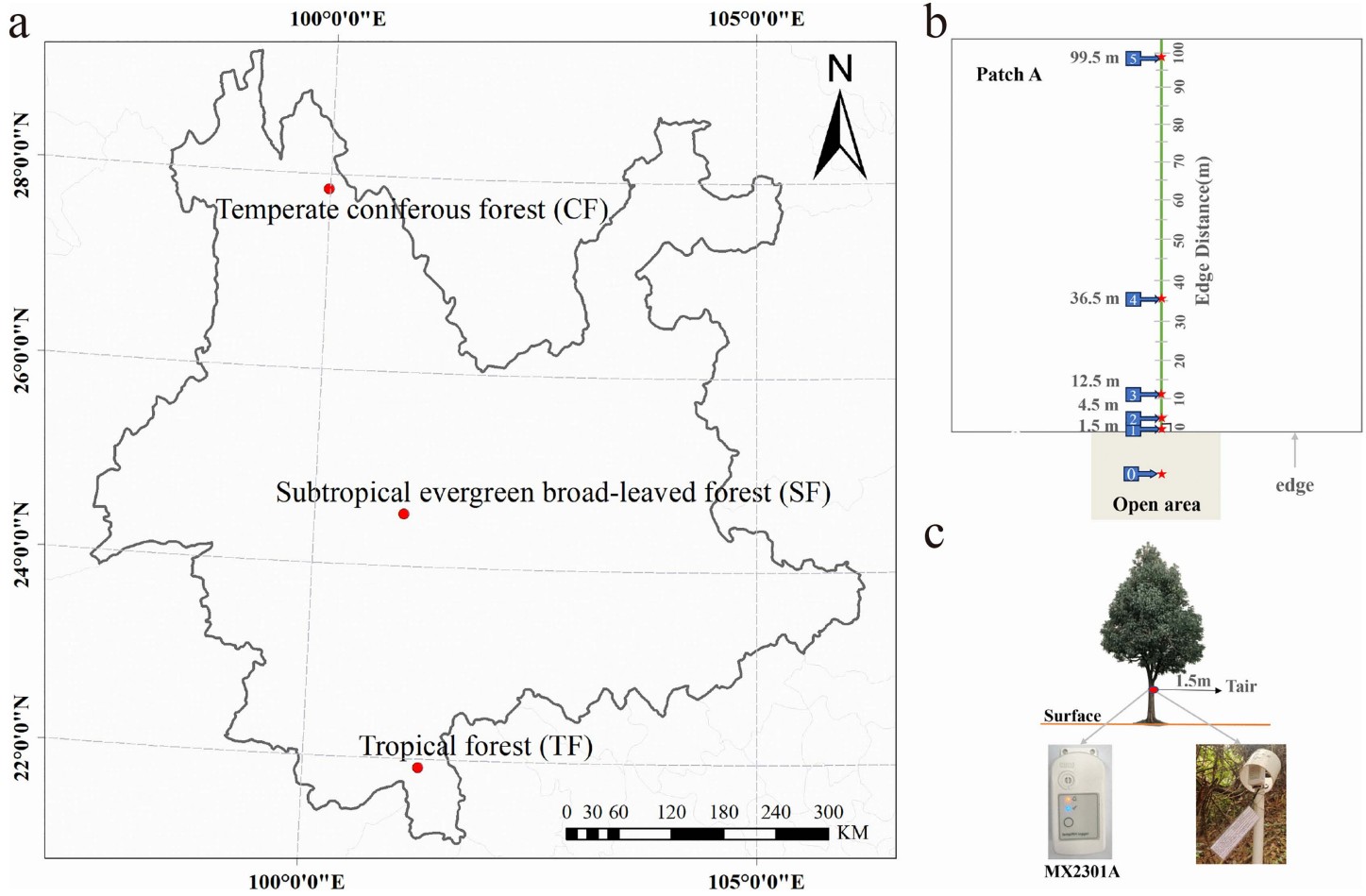

**Fig 1. Study region and monitoring design. (a)** Locations of the three study sites in Yunnan Province, southwestern China, representing temperate coniferous forest (CF), subtropical evergreen broadleaved forest (SF), and tropical forest (TF) along a climatic gradient. Provincial boundary data were obtained from Natural Earth (public domain; http://www.naturalearthdata.com/). **(b)** Schematic of the edge-to-interior transect design showing the open-area reference plot (plot 0) and forest plots at 1.5, 4.5, 12.5, 36.5, and 99.5 m from the edge. Each plot was a 3 m × 3 m quadrat. **(c)** Installation of the HOBO MX2301A logger used to measure air temperature at 1.5 m height. Photographs in panel (c) were taken by the authors.

## 2.2. Data collection

Transects were established at the boundary between forest and adjacent non-forest vegetation and were oriented toward the south. At each site, we selected a forest patch that allowed sampling to 99.5 m from the focal edge without encountering another edge. Because edge-to-interior microclimate gradients are often nonlinear and are frequently approximated by exponential decay [13,29,32], we implemented a clustered sampling design that concentrates measurements near the edge and increases spacing toward the interior. We established a 100 m transect in each forest, extending perpendicular to the edge into the interior. Following common edge-gradient designs [16,36], we located five forest plots at 1.5, 4.5, 12.5, 36.5, and 99.5 m from the edge. Each plot consisted of a 3 m × 3 m quadrat centered on the target distance, with a PVC pole at least 2m tall fixed to the ground. This layout increases measurement density where the gradient is expected to be steepest while retaining an interior anchor point for model fitting [16]. Sampling locations were also constrained by logger availability and maintenance logistics, and distance effects were evaluated using a continuous log-transformed

distance predictor and a DEI defined from interior confidence bounds as described in Section 2.3.3, rather than by a breakpoint tied to any single sampling location.

Air temperature was recorded using HOBO MX2301A loggers mounted at 1.5m height on each pole and shielded from direct radiation using a PVC cover with a diameter of 100 mm and a height of 10 cm [37]. At each site, an identical logger was installed in the adjacent open area at least 100 m from the forest edge to provide an open-reference series for computing forest–open temperature offsets (Fig 1b). Loggers recorded at 10 min intervals from May 2023 to May 2024, and data were retrieved wirelessly using Bluetooth Low Energy. Each logger produced 52,560 observations over the study year. Across three forests and five forest distances per site, we obtained 15 forest time series paired with their site-specific open-reference series, yielding 788,400 offset records prior to temporal aggregation.

To assess whether the year-long automated transect captured spatially representative edge-to-interior patterns within each forest, we established four additional transects parallel to the primary transect for snapshot measurements. This resulted in five transects per forest when including the original transect. In addition, we conducted stand-structure measurements and took hemispherical fisheye photographs in all plots across the five transects to derive forest structural metrics, such as canopy openness (Openness) and leaf area index (LAI), and the snapshot design and methodological details are described in the Supplementary Material.

## 2.3. Data analysis

**2.3.1. Temperature data preprocessing.** Temperature offset was calculated as $T_{\text{offset}} = T_i - T_0$, where $T_i$ is air temperature measured inside the forest at one of the five quadrats, and $T_i$ is air temperature measured in the adjacent open reference quadrat. Negative values indicate cooler conditions inside the forest relative to the open area, whereas positive values indicate warmer conditions.

To remove minor elevation differences within each transect, we first adjusted all temperature records to the elevation of the open reference quadrat using a standard lapse rate correction (−0.6°C per 100 m elevation gain), and then calculated $T_{\text{offset}}$ from the corrected temperatures. Seasonal analyses followed the Climate Season Division standard [38]. We defined spring as March to May, summer as June to August, autumn as September to November, and winter as December to February. A season was retained only when more than half of the temperature records for that season were available.

We determined daily sunrise, sunset, and solar noon times for each site using site latitude, longitude, elevation, and the monitoring period. Calculations were performed in R version 4.3.1 using the functions maptools::sunriset() and maptools::solarnoon(). All solar times were expressed in local time for Asia/Shanghai. Because sunrise and sunset vary little within a given month at each site, we calculated these times on a monthly basis and applied them to classify observations within that month.

We processed the temperature series in four steps. First, we aggregated 10 min records to hourly means using six records per hour, and any hour with missing 10 min observations was treated as missing. Second, we classified each hourly record as daytime or nighttime using the site-specific monthly sunrise and sunset times. Third, for each plot and day, we computed daytime hourly $T_{\text{offset}}$ values and defined daily MCI as the 5th percentile of daytime hourly offsets. This lower-tail statistic represents the strongest cooling because offsets are typically negative under canopy buffering, and it provides a robust estimate that is less sensitive to single-hour noise than the minimum. Fourth, for DEI estimation we computed the 5th percentile of daytime hourly air temperatures for each forest plot and for the open reference series on each day. These lower-tail temperatures, rather than offsets, were used to derive daily buffering slopes as described in Section 2.3.3, which aligns DEI estimation with the same cooling-relevant portion of the temperature distribution used for MCI.

**2.3.2. Temperature gradient from edge to interior and its temporal dynamics.** Edge-to-interior variation in MCI was analysed using linear mixed-effects models fitted in R version 4.3.1 with the lmer function in the lme4 package. Daily MCI values were modelled as a function of edge distance, forest type, and their interaction to test whether the strength of the edge gradient differed among forests. Edge distance was natural-log transformed to accommodate the expected

 

nonlinear change in microclimate along the transect. Alternative random-intercept structures were evaluated to account for repeated measurements and site-level heterogeneity. Candidate structures included plot, date, and date nested within site. Models were compared using AIC and the most parsimonious structure was selected. The final model retained a random intercept for date nested within site, which accounts for repeated daily observations while capturing site-specific day-to-day variation [16]. All fixed effects were retained in the final model regardless of statistical significance to ensure consistent inference on gradient terms and forest-type contrasts across temporal aggregations.

Temporal dependence in the edge gradient was assessed by fitting the same fixed-effect structure at multiple time scales. Models were applied to annual data using daily observations for each plot, and they were refitted to seasonal sub-sets with 90–93 daily observations per plot and to monthly subsets with 30–31 daily observations per plot. This approach allowed direct comparison of edge-gradient strength and forest-type differences among months, seasons, and the full year.

Spatial representativeness of the automated transect was evaluated using snapshot transects. The log-distance mixed model was refitted to the combined dataset that included both the year-long automated transect and snapshot measurements. This fit was used to generate predicted edge-to-interior response curves and corresponding 95% confidence intervals for each site (S4 Fig in S1 Data). Snapshot transects were collected in a different year and within a limited measurement window, so they were used only to assess spatial robustness and were not used to infer temporal dynamics.

**2.3.3. Quantifying the distance of edge influence (DEI).** The DEI was defined as the distance from the forest edge at which the predicted buffering metric first became statistically indistinguishable from interior reference conditions, based on the 95% confidence interval of the interior reference [2,39]. Interior conditions were approximated using the farthest plot at 99.5 m. For each forest and temporal window, the estimated buffering metric at 99.5 m and its 95% confidence interval served as the interior reference. Because buffering metrics were derived from lower-tail daytime temperatures, DEI describes edge influence on the cool end of the daytime temperature distribution rather than on high-temperature extremes.

Distance-specific daily buffering slopes were estimated in two stages. First, for each day and each forest plot, the 5th percentile of daytime hourly air temperature was calculated for the forest plot and for the adjacent open reference. Within each temporal window, forest values were regressed against the open reference for each distance point according to

$$T^{5\%}_{forest} = a + bT^{5\%}_{open} + \varepsilon$$

(1)

where b represents the coupling or buffering slope for that distance in the given time window. A slope of 1 indicates no buffering, values below 1 indicate cooling relative to the open area, and values above 1 indicate warming [2,40]. Use of identical sensors in the forest and in the open reference reduced potential instrument-related bias in the estimated coupling. Second, the distance-specific slopes were regressed against natural-log-transformed edge distance to obtain a fitted slope profile along the edge-to-interior gradient for each forest and temporal window. DEI was identified as the smallest distance at which the fitted slope entered the 95% confidence interval of the interior reference at 99.5 m. When the fitted slope did not enter the interior interval within the sampled transect, DEI was reported as not detectable for that time window.

**2.3.4. Comparing MCI between coupled and decoupled zones.** Coupled and decoupled thermal zones were delineated for each forest and temporal window using the corresponding DEI estimate. Forest plots with edge distances less than or equal to DEI were classified as coupled to the open-area reference, whereas plots located beyond DEI were classified as thermally decoupled. When DEI fell between two sampled distances, plots were assigned according to this distance rule. When DEI was not detectable within the 99.5 m transect, a decoupled zone was not defined for that window and zone-based comparisons were not performed.

Differences in MCI were evaluated using Kruskal–Wallis nonparametric tests. Within each forest type and time window, MCI distributions were compared between coupled and decoupled zones. Differences among forest types were assessed

separately within each zone. Post hoc pairwise comparisons were conducted using Dunn's test with Holm adjustment to control multiple testing. Group letters reflect the outcomes of these post hoc comparisons (Fig 4, S3 Fig in S1 Data).

## 3. Results

### 3.1. Temporal and spatial patterns of temperature changes from the edge to the interior

Maximum cooling intensity varied systematically with distance from the forest edge and differed among sites along the climatic gradient (Fig 2, Table 1). Each forest type was represented by a single focal site, so patterns are described as evidence from this climatic gradient rather than as definitive forest type effects.

Across the transect, the tropical site generally exhibited the most negative MCI values, indicating the strongest cooling relative to the open reference. The subtropical site was typically intermediate and the temperate coniferous site showed the weakest cooling (Fig 2, S2 Fig in S1 Data). This ordering was evident at annual and seasonal aggregations, where seasonal MCI in the subtropical site remained less negative than in the tropical site but more negative than in the temperate site (Fig 2, Table 1).

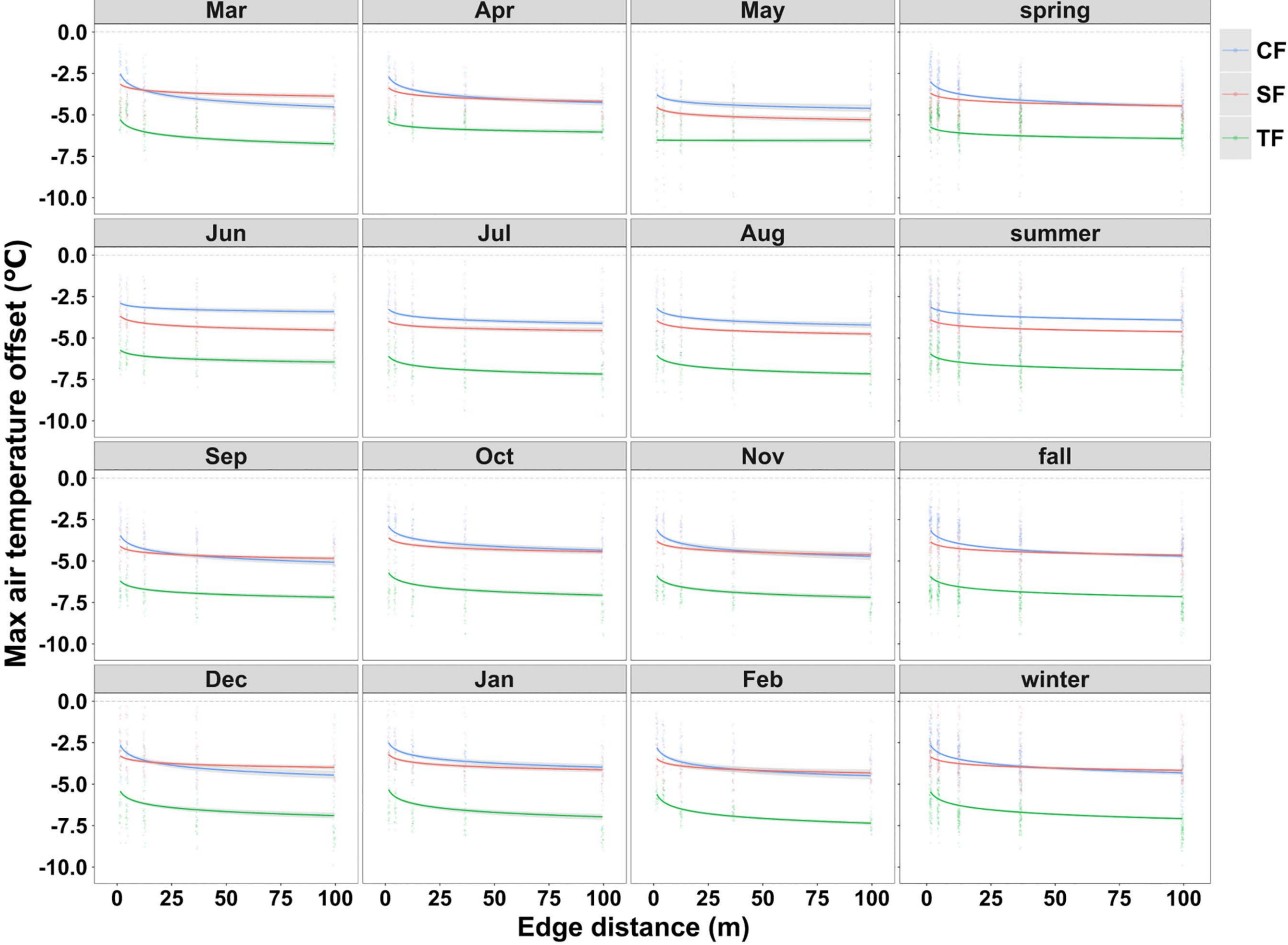

**Fig 2. Predictions of maximum cooling intensity as a function of edge distance at monthly and seasonal scales.** MCI is defined as the 5th percentile of daytime hourly temperature offsets derived from 10 min records, and more negative values indicate stronger cooling. Lines show model predictions for each forest site and shaded bands show 95% confidence intervals. Points show daily values for each distance along the transect.

**Table 1. Fixed effect estimates from log distance mixed effects models describing variation in maximum cooling intensity across edge distance, forest site, and their interaction at annual, seasonal, and monthly aggregations.**

| Max Air temperature offset (°C) | Exponential decay coefficient | | | Distance to edge (log-transformed, m) | | |
|---|---|---|---|---|---|---|
| | CF | SF | TF | CF | SF | TF |
| Whole year | 0.36** | −1.83*** | −1.04*** | −0.06 | −0.09** | −0.07. |
| Spring | 1.25*** | −4.24*** | −0.78* | −0.18. | −0.17* | −0.01 |
| Summer | 0.43. | −3.62*** | −2.12*** | −0.01 | −0.18*** | −0.06 |
| Fall | 0.99*** | −3.69*** | −2.67*** | −0.19** | −0.19*** | −0.09* |
| Winter | −0.38 | −2.89*** | −2.66*** | −0.19** | −0.20*** | −0.17* |
| Jan | 0.29 | −2.82*** | −2.85*** | −0.12 | −0.23** | −0.15. |
| Feb | 1.58*** | −3.53*** | −1.59*** | −0.22. | −0.20* | −0.22. |
| Mar | 2.43*** | −4.30*** | −0.21 | −0.34** | −0.15. | −0.22. |
| Apr | 1.36*** | −3.74*** | −1.31*** | −0.19* | −0.19** | 0.03 |
| May | −0.10 | −4.66*** | 0.84. | 0.01 | −0.18* | 0.15 |
| Jun | 0.65* | −3.58*** | −2.04*** | 0.08 | −0.20*** | 0.03 |
| Jul | 0.28 | −3.57*** | −2.56*** | −0.06 | −0.14** | −0.11. |
| Aug | 0.36 | −3.71*** | −1.77*** | −0.04 | −0.20*** | −0.08 |
| Sep | 1.00** | −4.02*** | −2.33*** | −0.22** | −0.17*** | −0.06 |
| Oct | 1.26*** | −3.34*** | −3.02*** | −0.15** | −0.20*** | −0.10* |
| Nov | 0.51 | −3.54*** | −2.84*** | −0.18** | −0.20*** | −0.10. |
| Dec | −0.64. | −2.35*** | −3.48*** | −0.24** | −0.18*** | −0.15* |

Subtropical forests (SF) were used as the reference forest ecosystem. The coefficient estimates of the models are given and the significance of the effect is indicated with asterisks (*=p<0.05, **=p<0.01, ***=p<0.001).

Monthly analyses revealed additional structure that was not captured by seasonal summaries. In March, September, and December, predicted cooling beyond 12.5 m was slightly weaker in the subtropical site than in the temperate site, indicating that monthly conditions can modify the apparent ordering among sites (Fig 2). Variation among months affected both the overall magnitude of MCI and the strength of the edge to interior gradient. The edge gradient was often steepest in the subtropical site, whereas the tropical site maintained the most negative offsets across distances in most months (Fig 2, Table 1). Daily time series further showed pronounced temporal dynamics in MCI across sites and distances (S1 Fig in S1 Data).

Snapshot derived offsets aligned with the automated transect predictions and remained within the 95% confidence bands across sites (S4 Fig in S1 Data). Supplementary structure analyses suggested that the distance response can be modulated by canopy structure in a site-specific manner, with significant negative interactions between log distance and LAI in the temperate and subtropical sites and between log distance and canopy openness in the temperate site (S1 Table in S1 Data).

### 3.2. The spatiotemporal variation of DEI

Thermal coupling between forests and the adjacent open reference weakened with increasing distance from the edge, as indicated by declining buffering slopes along the transects. Both the slope profiles and the resulting DEI estimates varied strongly through time at monthly and seasonal aggregations (Fig 3, Table 2).

At the monthly scale, DEI ranged from 17 to 59 m in TF when months with undetectable DEI were excluded. It ranged from 13 to 65 m in SF and from 18 to 77 m in CF. All sites showed an early summer contraction in DEI. The minimum occurred in May for CF and in June for both SF and TF (Fig 3, Table 2). Seasonal aggregation yielded DEI

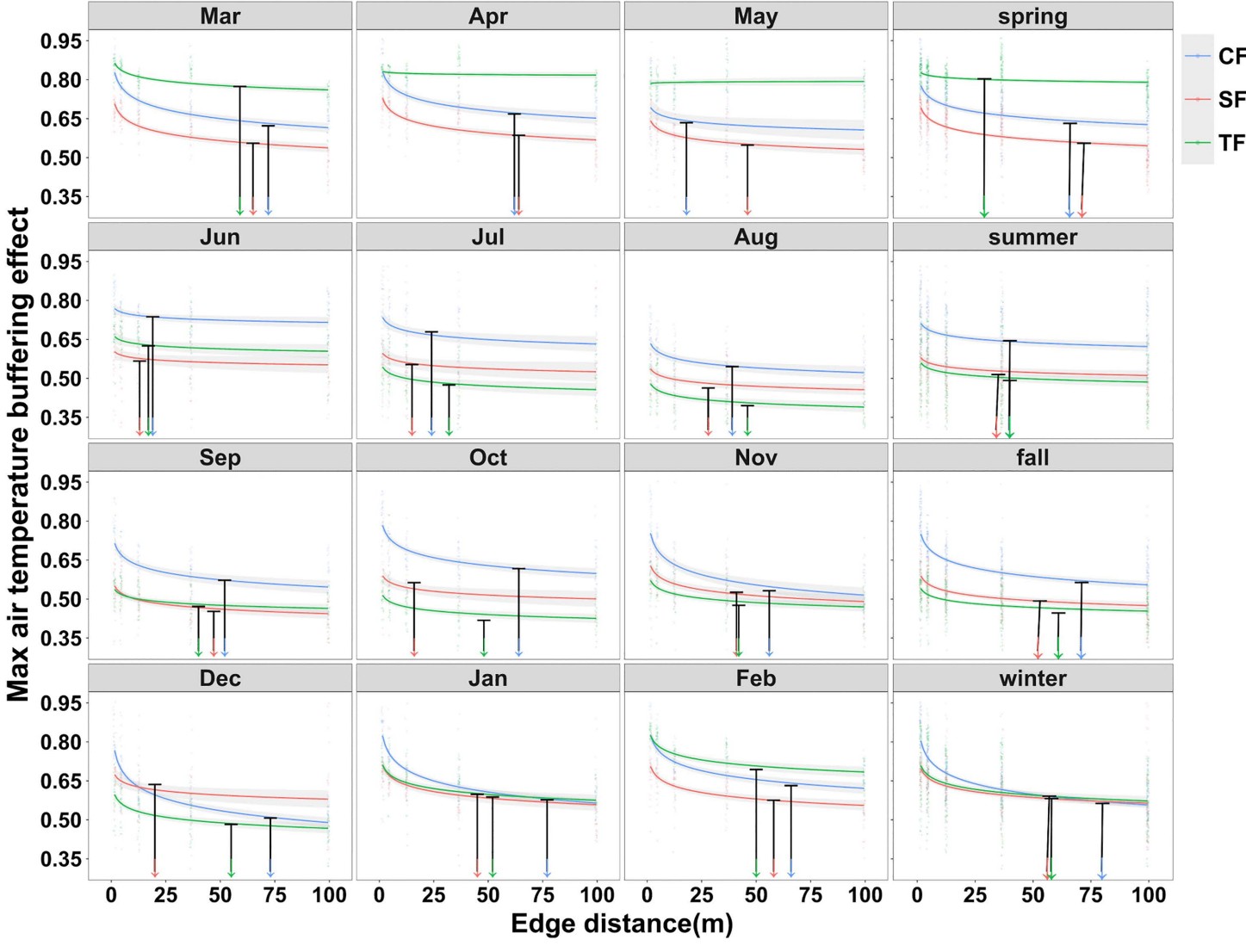

**Fig 3. Predicted buffering slopes along the edge to interior gradient at monthly and seasonal scales.** Slopes were derived from regressions of lower tail daytime air temperatures in forest plots against the adjacent open reference. Values of the slope below 1 indicate buffering, and smaller values indicate stronger decoupling. Lines show fitted relationships with log transformed edge distance and shaded bands show 95% confidence intervals. Black markers indicate DEI, defined as the smallest distance at which the fitted slope enters the 95% confidence interval of the interior reference at 99.5 m.

values from 29 to 61 m in TF, from 35 to 72 m in SF, and from 40 to 80 m in CF. At the annual scale, DEI was 56 m in TF, 72 m in SF, and 82 m in CF, which produced an ordering of TF below SF and below CF. This ordering did not consistently hold at monthly and seasonal scales, indicating that DEI depends on the temporal resolution used for estimation.

DEI could not be identified within 99.5 m in TF during April and May (Fig 3, S3 Fig in S1 Data). This outcome indicates that the fitted slope profile did not converge to the interior reference within the sampled transect during those months, which is consistent with either a DEI exceeding 99.5 m or a weakly defined interior threshold within the sampling extent.

**Table 2. Spatiotemporal variation in DEI across forest sites at monthly, seasonal, and annual scales.**

| Period | CF | SF | TF |
|---|---|---|---|
| Mar | 72(−4.2) | 65(−5.8) | 59(−6.3) |
| Apr | 62(−4.6) | 64(−4.9) | —— |
| May | 18(−5.4) | 46(−6.1) | —— |
| Jun | 19(−4.3) | 13(−5.9) | 17(−6.9) |
| Jul | 24(−5.2) | 15(−7.0) | 32(−7.5) |
| Aug | 39(−5.7) | 28(−6.3) | 46(−7.7) |
| Sep | 52(−5.7) | 47(−5.7) | 40(−7.8) |
| Oct | 64(−4.0) | 16(−5.5) | 48(−8.2) |
| Nov | 56(−5.8) | 41(−5.7) | 42(−7.6) |
| Dec | 73(−5.2) | 20(−4.2) | 55(−7.9) |
| Jan | 77(−4.5) | 45(−4.7) | 52(−7.7) |
| Feb | 66(−4.2) | 58(−5.3) | 50(−7.0) |
| Spring | 66(−4.6) | 72(−5.7) | 29(−6.3) |
| Summer | 40(−5.4) | 35(−6.5) | 40(−7.6) |
| Fall | 71(−5.2) | 53(−5.7) | 61(−7.9) |
| Winter | 80(−4.8) | 57(−4.8) | 58(−7.6) |
| Whole year | 82(−5.0) | 72(−5.7) | 56(−7.5) |

Values in parentheses report the median daily MCI within the decoupled interior for the corresponding time window and are presented for completeness. Months with undetectable DEI within 99.5 m are indicated by dashes.

### 3.3. Spatiotemporal variation in MCI within coupled and decoupled zones

Coupled edge and decoupled interior zones were delineated for each forest and time window using the corresponding DEI estimate. Plots with edge distances less than or equal to DEI were classified as coupled to the open reference, whereas plots located beyond DEI were classified as decoupled. Comparisons were restricted to months and seasons for which DEI was detectable within the 99.5 m transect.

Across temporal aggregations, decoupled interiors generally exhibited stronger extreme cooling than coupled edge zones, as indicated by more negative MCI values (Fig 4, S3 Fig in S1 Data). This contrast was evident at annual and seasonal scales and became more pronounced at the monthly scale, which revealed substantial within year variation in both the magnitude of extreme cooling and the difference between zones (Table 2, S3 Fig in S1 Data).

Cooling intensity also differed among sites along the climatic gradient. The tropical site showed the strongest extreme cooling in both zones, the subtropical site was typically intermediate, and the temperate coniferous site showed the weakest cooling (Fig 4). Monthly peaks in decoupled interior cooling occurred in different parts of the year among sites. Median decoupled zone MCI reached −8.2°C in October in the tropical forest, −7.0°C in July in the subtropical forest, and −5.8°C in November in the temperate coniferous forest (Table 2). These patterns indicate that the thermal benefit of decoupled interiors is not constant through time and that its seasonal timing differs across the climatic gradient. For April and May in the tropical forest, a decoupled zone was not defined because DEI was not detectable within the transect.

## 4. Discussion

Forest-edge microclimate buffering varied along two complementary dimensions. One dimension was the intensity of extreme cooling captured by MCI, and the other was the spatial extent of edge influence captured by DEI. Along the climatic gradient, the tropical site showed the strongest cooling, whereas the temperate coniferous site showed weaker

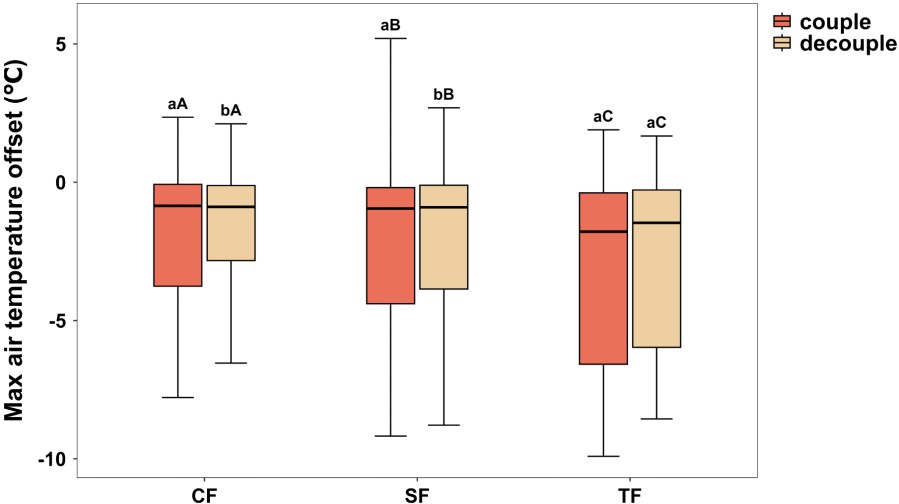

**Fig 4. Maximum cooling intensity in coupled edge and decoupled interior zones across the climatic gradient.** Zones were defined relative to DEI for each forest and time window, and comparisons were limited to windows with detectable DEI. MCI represents the lower tail extreme of daytime temperature offsets, and more negative values indicate stronger cooling relative to the open reference. Boxplots show the median and interquartile range, and whiskers show the full range. Lowercase letters indicate significant differences between coupled and decoupled zones within a forest site, and uppercase letters indicate significant differences among forest sites within a zone.

cooling but the largest annual DEI, with the subtropical site generally intermediate. Both metrics varied markedly among months, which indicates that thermally decoupled interiors expand and contract through the year and that microrefugia availability is strongly seasonal.

### 4.1. Temperature variation at the forest edge and maximum cooling

Across the three sites, maximum cooling strengthened from the forest edge toward the interior and approached an interior plateau, which indicates that the strongest thermal buffering relative to the open reference is typically observed away from the edge. The magnitude of this extreme cooling and the rate at which it accumulated with distance were not constant across the climatic gradient or across months. These patterns show that edge thermal buffering cannot be treated as a fixed property of a forest patch [2,16,41]. It emerges from interactions between background climate, canopy mediated energy exchange, and local structure that jointly shape the edge to interior gradient [12,13,36,42,43].

Differences in maximum cooling among sites are consistent with a heat load perspective. Warmer and more radiatively forced conditions can increase the temperature contrast between open areas and shaded forest air, especially during daytime hours when the open reference warms rapidly. Under such conditions, canopy shading reduces shortwave inputs to the understory, and evapotranspiration can further suppress air temperatures when water is not limiting [13,44,45]. Global and regional syntheses have shown that canopy structure and macroclimate jointly regulate understory temperature offsets, and that buffering tends to be stronger under higher ambient temperatures and in stands with greater canopy development and structural complexity [13,46,47]. Within this framework, stronger cooling at the tropical site and intermediate cooling at the subtropical site likely reflect a combination of higher external heat load and canopy functioning that can sustain cooling in the understory.

Seasonal timing also mattered, and the strongest decoupled interior cooling occurred in different months across sites (Table 2). This divergence in timing suggests that extreme buffering is favoured when strong radiative forcing coincides with conditions that maintain canopy and understory cooling capacity [12,45,48]. In monsoon influenced systems, for

example, periods with reduced cloud cover can increase daytime heating in the open reference, while antecedent rainfall can sustain evaporative cooling beneath the canopy [49,50]. In contrast, months with persistent cloudiness can reduce radiative contrasts between forest and open areas, which can weaken extreme offsets even when mean conditions remain different [12,44]. These mechanisms provide a parsimonious explanation for the observed month to month variability without requiring the assumption that forest type alone determines cooling intensity.

Monthly analyses further revealed that site ordering was not fully stable across the transect. Several months showed convergence or partial reversals between the subtropical and temperate sites at distances beyond 12.5 m, which indicates that synoptic conditions and seasonal shifts in canopy functioning can temporarily outweigh the average climatic ordering. Such episodes are easily masked by seasonal aggregation, and they emphasise that interpretations based on annual or seasonal means may understate the dynamism of edge buffering relevant to short lived heat stress [29,51–53].

Supplementary structure analyses suggest that local variation in canopy structure may influence how rapidly maximum cooling strengthens from the edge to the interior (S1 Table in S1 Data). These results suggest that spatial changes in canopy density and openness along the transect can modulate how rapidly cooling strengthens from the edge to the interior [16,36,43,54]. Denser canopies can reduce radiative inputs and damp air exchange with the surrounding matrix, which promotes faster accumulation of cooling with distance [43]. More open edges can increase shortwave penetration and turbulent mixing, which weakens cooling near the boundary and delays the emergence of interior like conditions [16,29,55]. Although these structural signals were forest specific, they align with established mechanisms linking canopy structure and mixing to microclimate buffering [36,55,56].

## 4.2. The Distance of Edge Influence (DEI) of forest temperature

DEI captures the spatial reach of edge influence by identifying where forest temperatures become statistically indistinguishable from an interior reference based on the coupling slope in Fig 3. Because DEI is defined relative to an interior confidence interval and because coupling can vary strongly through the year, the extent of thermally decoupled interiors should not be treated as static [15,57]. Instead, interior conditions can expand and contract seasonally, which implies that microrefugia availability is inherently time dependent [51,58,59].

At the annual scale, DEI increased along the climatic gradient, with the smallest value in TF and the largest value in CF. This ordering did not persist at shorter temporal aggregations, which highlights a scale dependence that is rarely evaluated in edge microclimate studies. Monthly minima occurred in early summer, with the minimum in May for CF and in June for both SF and TF (Fig 3, Table 2). Seasonal interpretation of DEI benefits from considering cooling intensity together with coupling. A smaller DEI indicates that the slope-based criterion for interior conditions is met closer to the edge, but this outcome can arise through more than one pathway [57,60]. Stronger buffering near the edge can reduce coupling more rapidly with distance, while increased coupling within the interior can move the interior reference closer to edge conditions and shorten the apparent DEI [15,29]. This ambiguity is intrinsic to any threshold-based definition and reinforces the value of examining DEI alongside MCI rather than interpreting DEI in isolation [57,60].

Reported DEI values from other regions span a wide range, and direct comparison is often limited by differences in definitions and estimation procedures. Typical DEI ranges have been suggested for selected biomes, including values on the order of 20 meters in tropical forests and values around or above 12.5m in temperate forests [2,16]. The broader range observed here likely reflects a combination of methodological and site-context effects. DEI is sensitive to how the interior reference is defined and to the width of the associated confidence interval [2,39]. Edge contrast, edge age, and edge history can also modulate the depth of edge influence and can therefore shift DEI even under similar climates [61–63]. In mountainous landscapes, topography can further shape air mixing and thermal gradients, which may alter how rapidly edge influence attenuates with distance [17,34,64–66].

Mechanistic attribution remains tentative because radiation, wind, humidity, and soil moisture were not measured directly. Seasonal shifts in energy balance and water availability nevertheless provide a coherent explanation for the

observed time dependence [12,44]. Periods with sufficient water availability and strong canopy shading can enhance latent heat flux and reduce radiative inputs to the understory, which would promote faster emergence of interior-like conditions and reduce DEI [13,16,67]. In contrast, in TF during April–May, edge–interior coupling appears to be especially strong, such that DEI may extend beyond the sampled 99.5 m transect. This pattern is consistent with reduced canopy cooling capacity and/or enhanced coupling to the surrounding matrix [11,68,69], which could shift DEI beyond the sampled extent. This pattern suggests that the spatial extent of decoupled interiors can be smallest when heat stress is potentially most acute, whereas stronger cooling later in the year can coincide with conditions that sustain canopy-mediated cooling [29,36,44]. Monthly assessment therefore provides information that cannot be recovered from annual estimates and it clarifies when and where edge influence is most likely to penetrate deeply into forest patches.

### 4.3. DEI-defined decoupled interiors as microrefugia and management implications

DEI-based delineation of coupled and decoupled zones provides a practical way to define microrefugia using thermal decoupling rather than a fixed distance alone. Across sites, decoupled interiors consistently exhibited stronger extreme cooling than coupled edge zones, which supports their functional interpretation as daytime thermal refugia under high heat load [58,59]. Cooling magnitudes in the decoupled interior were substantial and reached approximately 8°C in the tropical site. Such differences are large enough to shift thermal safety margins for temperature-sensitive understory organisms and to alter the conditions experienced during short-lived heat events [68,69].

Microrefugia defined in this way are inherently dynamic. Both the extent and, in some months, even the presence of thermally decoupled interiors varied markedly through the year, indicating that refugial availability is time dependent. This pattern is consistent with a seasonal contraction of thermally decoupled interiors or with a DEI that exceeded the sampled transect [29,57]. Either interpretation implies reduced refuge availability during a period that can coincide with high thermal stress in the pre-monsoon season. Seasonal variability in canopy energy exchange offers a coherent explanation. Shifts in cloudiness, evaporative demand, and water availability can change the temperature contrast between open and forest environments and can modify how rapidly interior-like conditions emerge with distance from the edge [12,44,70].

This decoupling framework has direct management relevance because it translates microrefugia into a measurable spatial threshold. A forest patch can only provide a DEI-defined refugium if it contains area where the distance to the nearest non-forest edge exceeds DEI. When the maximum distance to edge within a patch is smaller than the relevant DEI, a thermally decoupled interior is unlikely to form, and the patch may fail to sustain the strongest cooling function even if mean offsets remain negative. This provides a clear basis for evaluating fragmentation impacts and for prioritising patch retention [35,71]. Distance-to-edge maps can be combined with DEI estimates to quantify the fraction of each patch that qualifies as potential refugial core under specific months or seasons [72].

Conservative design for year-round buffering can be based on the largest DEI observed across seasons or across critical months, whereas climate-adaptive planning can target DEI values for the periods when heat stress risk is highest. This time-specific approach avoids relying on a single annual DEI that may obscure months when decoupled interiors contract [51,52]. Because each forest type was represented by one focal site, the numerical thresholds reported here should be interpreted as evidence from a climatic gradient rather than universal limits. Replication across multiple sites per forest type and concurrent measurements of radiation, wind, and moisture would further strengthen mechanistic attribution and improve transferability of DEI-based guidance for edge-aware conservation planning in fragmented landscapes [11,13,73].

### 4.4. Limitations and recommendations for future research

A primary limitation is that each forest type was represented by a single focal site and a single main transect. Year-long monitoring and consistent instrumentation improve internal comparability, but the design necessarily limits spatial representativeness and the ability to separate macroclimate effects from site-specific structure and topography [11,74]. The sampled stands were mature and relatively undisturbed, so inference is most applicable to long-unharvested forests

without recent stand-replacing events and should not be extended to regenerating stands, recently disturbed forests, or heavily managed edges without additional validation [16,33,75]. The interior reference was approximated by the farthest distance within a 99.5 m transect. Periods when edge influence extends beyond the sampled extent can therefore yield undetectable DEI or truncated estimates, which should be considered when interpreting time windows with weak convergence to interior conditions [29,57].

Mechanistic attribution is also constrained by the set of measured variables. Air temperature was monitored continuously, but radiation, humidity, wind, and soil moisture were not measured concurrently. Seasonal driver interpretations are therefore best viewed as hypotheses that are consistent with energy-balance principles rather than direct process tests [13]. Snapshot transects were used to evaluate spatial robustness of the distance response within sites, and their different year and limited sampling window prevent their use for diagnosing month-specific mechanisms. In addition, daily MCI values are temporally autocorrelated and observations are nested within transects. Nonparametric comparisons between coupled and decoupled zones should therefore be interpreted as descriptive evidence of differences rather than as fully independent tests, and future work would benefit from mixed modelling or autocorrelation-aware resampling such as block bootstrap designs that respect date and plot structure [76–78].

Future studies should replicate transects within each forest type and across multiple sites, ideally spanning gradients in stand age, successional stage, and edge history [75,79–81]. Replication across edge orientations, slope positions, and elevation bands would allow a clearer evaluation of how terrain and prevailing flows interact with edge effects [82–85]. Expanded process measurements would strengthen mechanistic inference, particularly concurrent radiation, wind, humidity, and soil moisture observations that can link coupling and cooling intensity to changes in mixing and evaporative capacity [12,67]. Multi-year monitoring is also needed to quantify interannual variability in extreme events and to test whether DEI and extreme cooling are stable across contrasting years.

At broader scales, landscape configuration should be integrated explicitly. Patch size and shape determine the maximum distance to edge and thus the potential area that can qualify as a DEI-defined refugial core [72,86–88]. Combining DEI estimates with distance-to-edge mapping would enable spatially explicit evaluation of which patches can sustain thermally decoupled interiors during critical months. Finally, linking thermal gradients to species occurrence, habitat use, and functional diversity will clarify ecological consequences [84,89]. Some taxa may benefit from warmer edge environments, so weaker buffering does not necessarily imply degradation [90,91]. A combined view of cooling benefits, niche partitioning, and connectivity will help translate edge microclimate metrics into conservation strategies that are both species-specific and robust to uncertainty.

## 5. Conclusions

This study applied a consistent transect-based design and high-frequency air-temperature monitoring to quantify edge-to-interior thermal gradients in natural forest patches along a climatic gradient in southwestern China. Temperature offsets became increasingly negative with distance from the edge, indicating progressively stronger cooling toward forest interiors. Extreme cooling intensity and the spatial reach of edge influence were not constant through time. Both maximum cooling intensity and DEI varied substantially among months and differed among sites along the climatic gradient. Peak monthly median cooling in the decoupled interior reached 8.2°C in the tropical forest, 7.0°C in the subtropical forest, and 5.8°C in the temperate coniferous forest.

A key contribution of this study is the explicit separation of intensity and extent as complementary dimensions of edge microclimate buffering. MCI quantifies the strength of extreme cooling, whereas DEI delineates the distance required for interior-like thermal decoupling from the surrounding open area. DEI also proved strongly dependent on temporal aggregation. Monthly DEI estimates could diverge markedly from seasonal and annual values within the same forest, and in some periods a decoupled interior was not detectable within the sampled transect. This scale dependence indicates that microrefugia availability expands and contracts seasonally and cannot be inferred reliably from a single annual estimate.

These findings translate into operational guidance for fragmented landscapes. A forest patch can only provide a DEI-defined thermal refugium when it contains area where the distance to the nearest non-forest edge exceeds the relevant DEI for the period of interest. Conservatively, buffer-width planning and patch retention can be informed by the largest DEI observed across critical months or seasons, while monthly estimates can identify periods when interior refugia are most limited. At the same time, warmer edge environments may also support taxa that exploit higher temperatures, light availability, or seasonal resources, so reduced buffering near edges should not be interpreted as universal ecological degradation.

Overall, the present study highlights that edge microclimate buffering is dynamic in both strength and spatial reach. Incorporating temporally explicit DEI and extreme cooling metrics into distance-to-edge mapping and connectivity assessments offers a practical pathway for climate-aware conservation planning. Broader replication across multiple sites per forest type and concurrent measurements of radiation, wind, and moisture will improve transferability and strengthen mechanistic attribution.

## Supporting information

**S1 Data.** S1 Fig. Spatiotemporal sequence chart of daily MCI (5th percentile of daytime hourly temperature offsets) in natural forest types. The lines show the daily variation trend of MCI. Different panels represent natural forest type in different climatic zones. The colors of the lines and dots show the distance from the edge. Slight jittering has been applied along the X-axis to improve clarity. S2 Fig in S1 Data. Predictions of robust maximum cooling (MCI, C) as function of the distance to the edge (m). The lines show model predictions of significant interaction between forest ecosystem types and edge distance. The colors of the lines and points show natural forest type in different climatic zones. Slight jittering has been applied along the X-axis to improve clarity. S3 Fig in S1 Data. Spatial and temporal comparison of robust maximum cooling (MCI) in the decoupled interior and coupled edge zones of natural forest types at monthly and seasonal scales. Figure a is monthly scale, figure b is seasonal scale. Different panels represent natural forest type in different climatic zones. Comparison of maximum cooling between coupled and decoupled zones is represented by lowercase letters, while differences among forest types across months and seasons are represented by uppercase letters. The horizontal line in a box plot represents the median of the data, while the box limits indicate the interquartile range, extending to the minimum and maximum values. S4 Fig in S1 Data. Model predictions of the air-temperature offset (C) as a function of distance from the forest edge (m), combining the original automatic monitoring data and the new snapshot-transect measurements. Colored lines indicate natural forest types in different climatic zones (CF = temperate coniferous forest, SF = subtropical evergreen broadleaf forest, TF = tropical forest). Gray shaded ribbons show 95% confidence intervals. Point transparency and jittering were applied to improve visibility. S1 Table. GAM results for stand structural variables in different forest types, including key linear interaction terms with log-distance and a distance-to-edge smooth on air-temperature offset (C). (ZIP)

## Acknowledgments

We are grateful for the support provided by the Xishuangbanna Station of Tropical Rainforest Ecosystem Studies (National Forest Ecosystem Research Station at Xishuangbanna), the Chinese Academy of Sciences, the Ailao Mountain Nature Reserve Ecological Station and the Shangri-La Pudacuo National Park Administration and Nature Reserve during our field work.

## Author contributions

**Conceptualization:** Zhangjian Xie, Bin Wang.

**Data curation:** Zhangjian Xie.

**Formal analysis:** Zhangjian Xie, Qifei Chen.

**Funding acquisition:** Zhiming Zhang.

**Investigation:** Zhangjian Xie, Qifei Chen, Wenjun Liu, Hong Wang, Yuxin Ma, Yajie Jiang, Wen Liu, Yufeng Ma.

**Methodology:** Zhangjian Xie, Bin Wang, Qifei Chen, Wenjun Liu.

**Project administration:** Zhiming Zhang.

**Resources:** Zhiming Zhang.

**Supervision:** Zhiming Zhang.

**Validation:** Zhangjian Xie.

**Visualization:** Zhangjian Xie, Qifei Chen, Yuxin Ma, Weihong Liu, Yajie Jiang, Wen Liu.

**Writing – original draft:** Zhangjian Xie.

**Writing – review & editing:** Zhangjian Xie, Bin Wang, Wenjun Liu, Hong Wang, Weihong Liu, Cameron Proctor, Hans J. De Boeck, Zhiming Zhang.

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
