## [Decision Letter · Decision Letter 0]

7 May 2025

Dear Dr. Wang,

Thank you for submitting your manuscript to PLOS ONE. After careful consideration, we feel that it has merit but does not fully meet PLOS ONE’s publication criteria as it currently stands. Therefore, we invite you to submit a revised version of the manuscript that addresses the points raised during the review process.

We look forward to receiving your revised manuscript.

Kind regards,

Lingye Yao, Ph.D.

Academic Editor

PLOS ONE

[This research is supported by the National Natural Science Foundation of China (32260291), and the Project for Talent and Platform of Science and Technology in Yunnan Province Science and Technology Department (202205AM070005), the Major Program for Basic Research Project of Yunnan Province (202101BC070002), the Key Research and Development Program of Yunnan Province (202303AC100009) and the Scientific Research Fund Project of Yunnan Education Department (2025Y0076). We are grateful for the support provided by the Xishuangbanna Station of Tropical Rainforest Ecosystem Studies (National Forest Ecosystem Research Station at Xishuangbanna), the Chinese Academy of Sciences, the Ailao Mountain Nature Reserve Ecological Station and the Shangri-La Pudacuo National Park Administration and Nature Reserve during our field work.]

[ZZM, 32260291, National Natural Science Foundation of China, https://www.nsfc.gov.cn/

ZZM, 202205AM070005, Project for Talent and Platform of Science and Technology in Yunnan Province Science and Technology Department, https://kjt.yn.gov.cn/

ZZM, 202101BC070002, Major Program for Basic Research Project of Yunnan Province, https://kjt.yn.gov.cn/

ZZM, 202303AC100009, Key Research and Development Program of Yunnan Province, https://kjt.yn.gov.cn/

MYX, 2025Y0076, Scientific Research Fund Project of Yunnan Education Department, https://jyt.yn.gov.cn/

The funders had no role in study design, data collection and analysis, decision to publish, or preparation of the manuscript.]

Reviewers' comments:

Reviewer's Responses to Questions

**Comments to the Author**

1. Is the manuscript technically sound, and do the data support the conclusions?

Reviewer #1: No

Reviewer #2: Partly

Reviewer #3: Partly

Reviewer #4: Partly

2. Has the statistical analysis been performed appropriately and rigorously?

Reviewer #1: Yes

Reviewer #2: Yes

Reviewer #3: No

Reviewer #4: Yes

3. Have the authors made all data underlying the findings in their manuscript fully available?

Reviewer #1: No

Reviewer #2: Yes

Reviewer #3: Yes

Reviewer #4: Yes

4. Is the manuscript presented in an intelligible fashion and written in standard English?

Reviewer #1: No

Reviewer #2: Yes

Reviewer #3: Yes

Reviewer #4: Yes

Reviewer #1: This is an interesting study that quantifies the cooling effect of three forest types in relation to edge effects. Despite its simplicity, the study offers valuable insights with broader ecological implications. That said, the manuscript has several shortcomings that need to be addressed to improve its overall quality. In its current form, I do not believe this paper is ready for publication (either here or elsewhere). However, I encourage the authors to revise and resubmit, as the topic holds promise.

General comments

The manuscript lacks a clear and logical structure, which significantly affects readability and continuity. Most importantly, the paper would greatly benefit from professional English language editing. As it stands, the manuscript is difficult to read, understand, and follow—from the abstract to the conclusions.

The inclusion of monthly dynamics and values in figures and tables needs justification. In my opinion, the seasonal values are the most relevant and should be emphasized. Monthly data could be moved to supplementary materials, especially since they are neither discussed nor integrated meaningfully in the main text.

The paper lacks complementary information that could help contextualize and better explain the observed temperature dynamics (see specific comments). Including such data or background would enhance the scientific value and interpretability of the results.

Specific comments

L59: The sentence “approximately 20% of global forest area being situated within 100 meters of forest edges” is awkwardly phrased.

L66: The term “more negative offset” appears here without prior explanation. However, it is only introduced and defined later in lines 155–156. This disrupts the logical flow of the text and can confuse readers. Consider introducing this concept earlier or restructuring the text accordingly.

L67-69: This sentence is awkwardly formulated.

L69-72: This sentence is also poorly constructed. Additionally, it lacks a clear explanation of how previous methods led to an underestimation of the cooling effects. A more detailed rationale would improve the reader’s understanding.

L80: The abbreviation “DEI” is used without explanation. It is only introduced later (lines 184–188).

L128 (Figure 1): The text in panel b is difficult to read. Please enhance the figure’s legibility. Also, avoid using “patch B” to refer to the non-forested area, use a more descriptive term like “open area.”

L133 (Section 2.2): This section lacks critical information needed to interpret the temperature patterns. The following variables should be included or at least discussed:

-Canopy cover values for each plot

-Leaf Area Index (LAI)

-Forest age (are the plots composed of young, mature, or old-growth forests?)

-Slope aspect (e.g., north-facing slopes receive more sun and tend to be drier and hotter, while south-facing slopes are generally cooler and moister)

Additional Considerations:

-The study focuses solely on horizontal distance to the forest edge. What about vertical buffer capacity? (see https://doi.org/10.5194/bg-17-6423-2020).

-Nighttime temperature patterns are not addressed. For instance, are forest interiors warmer at night during winter months? The author’s dataset includes this information, and it would be valuable to explore these dynamics.

L165: I understand the rationale for using the 5th percentile, but the method of selecting values for hourly and daily temperature offsets remains unclear. For hourly temperature offsets, does this mean you selected the lowest 5% of hourly temperature differences per day? And for daily offsets, did you use the lowest 5% of daily values across each month, season, or year? Please clarify this methodology, it’s currently ambiguous.

L167 (Section 2.3.2): The construction of the linear mixed-effects model needs to be more clearly explained. You refer to “transects within region,” but there is only one transect per region, correct? Also, “daily temperature measurements” implies a temporal structure, shouldn’t there be random intercepts for plot and date to account for spatial and temporal autocorrelation? The current description lacks sufficient detail to understand how the random effects structure was defined.

L195-197: This information has already been mentioned in lines 144–145. Rather than repeating it, consider integrating or complementing the earlier explanation.

L207-212: This section is somewhat disorganized and hard to follow, please revise it.

L218: The phrase “The maximum temperature offset at the edge” is vague. Are the authors referring to the offset at 0.5 m, 99.5 m, or over the entire transect? Please be precise.

L219: The reported values seem inconsistent with Figure 2. For example, extracting the May values from Figure 2 suggests higher offsets than those stated. Please verify the accuracy of these numbers or clarify any differences due to methodology.

L220-221: The claim “The maximum air cooling at the edge of subtropical forests is significantly lower than in tropical forests, yet significantly stronger than in temperate forests” does not seem fully supported by the data in Figure 2. The cooling effect in subtropical forests (SF) is only slightly higher than in temperate forests (CF) and is actually lower during winter. Please revise or nuance this statement to accurately reflect the data.

L223-225: The statement “cooling ability of subtropical forests decreases significantly with increasing distance from the edge” lacks clear support in Table 1. It’s unclear where this conclusion comes from.

L230 (Figure 2): Please add a detailed legend description to the figure caption for better interpretation. It would also be valuable to discuss why May is the only month where almost no differences were observed between forest types or distances.

Table 1: It’s unclear what value is added by presenting monthly results here if they are not discussed in the main text. Consider either integrating these findings into the discussion or moving this information to the supplementary materials.

L266-268: “In April and May, the temperature of TF remained coupled with the macroscopic climates” First, fix the sentence “macroscopic”, and second this can be seen in Figure 3. The following sentence “but the maximum cooling in decoupled areas between forest edges and the macroclimate was more stable compared to coupled areas” is confusing, I don’t understand what you mean here and where this is coming from.

L284 (Discussion section): This section contains a considerable amount of speculation. Many of the mechanisms proposed to explain the observed buffering or cooling effects were not studied directly in this work. This weakens the interpretive strength of the discussion and leaves the study feeling incomplete.

For instance: L301-303, L328, L337-339 and other similar statements suggest causal explanations that were not empirically tested. Consider clearly distinguishing between speculation and evidence-based conclusions and limit overinterpretation.

L309-311: This is the first instance where conflicting findings from other studies are attributed to potential measurement errors. Such a claim should be made cautiously and, if retained, needs to be well-supported with context or methodological comparisons. Avoid discrediting other work without strong justification.

L315-317: Yes, indeed, your focus on extreme temperatures may explain some differences in findings compared to previous studies.

Reviewer #2: I have attached a review document which describes in details my criticisms of the manuscript. Overall the manuscript is a nice contribution to forest science but there are some limitation that are of concern for publication.

Reviewer #3: Dear authors,

This study investigates the spatiotemporal variation of maximum cooling effects from the edge to the interior of natural forest patches across three climate zones in Yunnan Province, China. Using high-resolution, in situ temperature monitoring along forest transects, the authors quantified how cooling intensity and the Distance of Edge Influence (DEI) fluctuate by forest type, season, and month. The results reveal that forests in warmer regions exhibit stronger and more stable cooling effects, with substantial temporal dynamics in DEI and maximum temperature buffering capacity across forest types.

I found the topic and experimental design to be appropriate, but I had concerns about the analysis methods, and I noted that several parts of the manuscript lack sufficient explanation.

___ Major Issues __

(1) The definitions of "coupled" and "decoupled" zones in your study are based on patterns of temperature offset across edge distances, using DEI as a threshold. Given this, it may appear circular to then compare the magnitude of temperature offset between these zones as if they are independent. I recommend clarifying how the DEI threshold was determined (e.g., via regression residuals or confidence intervals) and whether this procedure is sufficiently orthogonal to the maximum cooling values being compared. This would strengthen the validity of the comparison in Figure 4 and help avoid concerns of tautological reasoning.

(2) In Section 2.3.4, the manuscript states that maximum cooling was quantified across seasonal, monthly, and annual time scales and even refers to high-resolution (e.g., hourly or daily) temperature data in the methods. However, the results section does not present findings on maximum cooling at hourly or daily temporal resolutions. If such analyses were indeed conducted, the results (or at least a summary thereof) should be explicitly included in the manuscript or supplementary materials. If not, please clarify this in the text to avoid confusion and ensure methodological transparency.

___ Minor Issues ___

(3) In line 108, the hypothesis that cooling and DEI are "more pronounced in forest ecosystems in warmer regions" appears somewhat abrupt. This important claim is not clearly supported by the preceding context in the introduction. I recommend adding a brief rationale earlier in the paragraph to explain why warmer climatic regions would be expected to show more substantial cooling effects or greater DEI-ideally referencing previous findings or theoretical justifications. This will help readers better understand the foundation of your hypothesis.

(4) Line 208:

Capitalize the first letter of this sentence.

(5) Line 273:

Remove the duplicated phrase "which occurs at the monthly scale."

Reviewer #4: It was a pleasure to review the manuscript “Spatiotemporal variation of the Maximum Cooling Effect across Edge-to-Interior Gradients in Natural Forest Patches of Southwest China” for PLOS One. In this manuscript, authors examined the spatiotemporal dynamics of maximum cooling at forest edges across three forest types in Southwest China. The study has strong relevance, particularly given increasing interest in how forest microclimates buffer against global climate change. Results clearly show that maximum cooling and DEI vary by forest type and season, with warmer forests exhibiting stronger cooling but smaller DEI. The findings are well contextualized with existing global studies. I think this work has potential for publication, especially as it tackles an understudied topic, edge-related microclimate variability but I will leave the decision on fit and scope to the editor. Especially since I am not an expert on this topic. So most of the issues I raised are statistical/inference concerns and I would recommend that this manuscript being peer reviewed by an expert on forest microclimates.

My main concern is with clarity around the methods and the framing of novelty. While the abstract mentions transect-based in situ monitoring, it lacks detail on spatial replication, sensor placement, or data resolution, which are critical to interpreting spatiotemporal patterns. Also, terms like “maximum cooling” need more precise definition—does this refer to mean maximum difference in temperature between edge and interior, or some peak event? Without these clarifications, the magnitude and implications of their findings are hard to judge.

The logic and novelty of comparing different forest types (temperate, subtropical, and tropical) are compelling, but the manuscript does not clearly articulate how these comparisons inform theory or practice beyond the descriptive level.

I like the flow of the manuscript, it has great coherence and cohesion, and it makes the readers enthusiastic about following the manuscript and continue reading it.

Once methodological clarity is improved and the authors better articulate their conceptual contributions, I would be happy to provide more detailed comments on a full version of the manuscript.

Main comments:

- Authors claim to support existing hypotheses, but they don't indicate what those hypotheses are. It would help to be more specific about how these findings advance or challenge current understanding.

- The manuscript could better characterize the nature of forest edges (e.g., abruptness, adjacent land use) and discuss how edge contrast might influence microclimate gradients.

- The study relies on a limited number of transects per forest type, which may restrict the generalizability of the findings. Future work should increase spatial replication to account for within-type variability.

- The authors state that they used a stepwise procedure to remove non-significant effects. Stepwise model selection (particularly automated approaches) is widely criticized for inflating Type I error, ignoring model uncertainty, and producing biased coefficient estimates. It’s often not recommended unless justified (e.g., via information-theoretic criteria like AIC/BIC and cross-validation). Therefore, authors should clarify whether this was based on AIC, BIC, or p-values, and ideally use model comparison techniques grounded in model selection theory (e.g., multimodel inference or model averaging).

- Daily temperature data was analyzed using linear mixed models, but there's no mention of controlling for temporal autocorrelation. Repeated daily measurements at the same locations can violate the assumption of independence, which may inflate Type I error or bias standard errors. Authors should consider including temporal autocorrelation structures or explicitly discuss how they addressed this issue.

-It’s stated that sensors “effectively avoid direct solar radiation”. This needs elaboration, such as what shielding was used? Were sensors inter-calibrated? What’s the margin of error?

Perhaps include calibration protocol, make/model of sensors, and specifications on shielding.

-Day/night was classified using monthly averages of sunrise/sunset. This may be coarse for fine-scale hourly temperature analyses, especially around dawn/dusk or in areas with significant elevation change. I think hourly classification using exact date/time (e.g., daily ephemerides) would improve accuracy.

-Statements such as 270-271 “this represents the upper limit of cooling that forests can offer in response to macroclimatic warming” may be overreaching without modeling future climate interactions or including multiple years of data.

-Figures are generally clear, but some (e.g., Table 2) are dense and could be reformatted for readability. Including schematic diagrams of the sampling design in the main text would aid comprehension.

-The discussion could briefly address potential implications under future climate change scenarios.

Minor comments:

Line 55: Leave the citations for the end of the sentence to improve readability.

Line 117-118: scientific names should be italicized.

Line 121: ditto

Line 140: You mentioned existing studies, but you only cited one study.

Line 271-274: Proof-read, repetitions included.

**Do you want your identity to be public for this peer review?** For information about this choice, including consent withdrawal, please see our Privacy Policy

Reviewer #1: No

Reviewer #2: **Yes:** Travis Heckford

Reviewer #3: No

Reviewer #4: **Yes:** Danial Nayeri

---

## [Author Response · Author response to Decision Letter 1]

27 Jul 2025

The responses to all reviewers have been provided in the attached file.

---

## [Decision Letter · Decision Letter 1]

20 Aug 2025

PLOS ONE

Dear Dr. Wang,

Thank you for submitting your manuscript to PLOS ONE. After careful consideration, we feel that it has merit but does not fully meet PLOS ONE’s publication criteria as it currently stands. Therefore, we invite you to submit a revised version of the manuscript that addresses the points raised during the review process.

**Comments from Academic Editor:**

Two of the four reviewers (Reviewer  #1 and #2) raise concerns about the data size, particularly regarding the sample size and temporal scale, which limits the generalizability and applicability of this study. Please address these concerns carefully considering the reviewers' comments for further and thorough improvements.

We look forward to receiving your revised manuscript.

Kind regards,

Lingye Yao, Ph.D.

Academic Editor

PLOS ONE

Journal Requirements:

Reviewers' comments:

Reviewer's Responses to Questions

**Comments to the Author**

Reviewer #1: All comments have been addressed

Reviewer #2: All comments have been addressed

Reviewer #3: All comments have been addressed

Reviewer #4: All comments have been addressed

2. Is the manuscript technically sound, and do the data support the conclusions?

Reviewer #1: Partly

Reviewer #2: Yes

Reviewer #3: Yes

Reviewer #4: (No Response)

3. Has the statistical analysis been performed appropriately and rigorously?

Reviewer #1: No

Reviewer #2: Yes

Reviewer #3: Yes

Reviewer #4: Yes

4. Have the authors made all data underlying the findings in their manuscript fully available?

Reviewer #1: Yes

Reviewer #2: Yes

Reviewer #3: Yes

Reviewer #4: Yes

5. Is the manuscript presented in an intelligible fashion and written in standard English?

Reviewer #1: Yes

Reviewer #2: Yes

Reviewer #3: Yes

Reviewer #4: Yes

Reviewer #1: I appreciate the effort you’ve put into revising this manuscript based on my previous comments. The manuscript is well-written, and the results on forest microclimate dynamics are interesting. However, I have concerns about the study’s robustness and generalizability that it can’t address in its current stage. The study’s reliance on a single transect per forest type and data from only one year severely limits its ability to capture spatiotemporal temperature variability. For sufficient statistical rigor, I recommend using at least three replicate transects per forest type and collecting data over multiple years. Without these, the findings lack the robustness needed for broader application. Additionally, the study overlooks critical forest structural characteristics, such as tree density, canopy height, and cover, which directly affect shading and cooling. A young forest with sparse canopy differs significantly from a mature, dense forest, and these differences aren’t addressed. As a result, the findings are too specific to the studied forests and cannot be extrapolated to other contexts, reducing the study’s impact. Given these limitations, the manuscript’s conclusions are not sufficiently supported for general applicability. I encourage you to redesign the study with replicate transects and multi-year data and to incorporate forest structural variables in future work. These changes are essential to elevate the study’s scientific contribution.

Reviewer #2: (No Response)

Reviewer #3: (No Response)

Reviewer #4: It was a pleasure to re-review the manuscript “Spatiotemporal variation of the Maximum Cooling Effect across Edge-to-Interior Gradients in Natural Forest Patches of Southwest China” for PLOS One.

As I mentioned earlier, I am not an expert in this particular context, but it was encouraging to see that three other reviewers with subject-matter expertise provided thoughtful feedback to strengthen the manuscript. I also appreciated seeing how respectfully and thoroughly the authors engaged with the critiques, incorporating the suggested edits and comments. I want to highlight and commend the authors for their responsiveness and effort in improving the draft.

The manuscript has improved significantly since the previous round, the language, structure, and overall flow are much clearer now. I believe the authors have thoroughly addressed all the comments I previously raised. However, as my expertise on this topic is limited, I’ll defer to the editor and the other reviewers, who have deeper knowledge in this area, to make the final assessment and recommendations.

**Do you want your identity to be public for this peer review?** For information about this choice, including consent withdrawal, please see our Privacy Policy

Reviewer #1: No

Reviewer #2: No

Reviewer #3: No

Reviewer #4: No

---

## [Author Response · Author response to Decision Letter 2]

30 Oct 2025

All the modification and explanatory documents have been uploaded as attachments.

---

## [Decision Letter · Decision Letter 2]

30 Nov 2025

Dear Dr. Wang,

We look forward to receiving your revised manuscript.

Kind regards,

Lingye Yao, Ph.D.

Academic Editor

PLOS ONE

Journal Requirements:

Reviewers' comments:

Reviewer's Responses to Questions

**Comments to the Author**

Reviewer #1: All comments have been addressed

Reviewer #3: All comments have been addressed

2. Is the manuscript technically sound, and do the data support the conclusions?

Reviewer #1: Yes

Reviewer #3: Yes

3. Has the statistical analysis been performed appropriately and rigorously?

Reviewer #1: Yes

Reviewer #3: Yes

4. Have the authors made all data underlying the findings in their manuscript fully available?

Reviewer #1: No

Reviewer #3: Yes

5. Is the manuscript presented in an intelligible fashion and written in standard English?

Reviewer #1: Yes

Reviewer #3: Yes

Reviewer #1: A few comments on authors' responses:

After reading the abstract and conclusions, the authors report the degrees of cooling effect for each forest type but fail to explain the mechanisms underlying why each forest exhibits that degree of cooling. After all, this is a research paper, not a report.

The authors do not sufficiently stress the practical implications of these forest edge-to-interior thermal dynamics for forest management and species adaptation in the Introduction and Discussion (implications that must be the main motivation of this study and a key strength). Elaborating further with specific ecological examples (e.g., effects on understory biodiversity) would broaden its appeal and strengthen the narrative.

The replication snapshot is not ideal but an interesting way to overcome the lack of replication. Please add details about the date and time period: Was it a few hours of measurement on a single day, one week, or spread over a month?

Forest structure characteristics add more value to the whole study. It is a pity that you missed measuring the most important variable: canopy cover/closure/openness, taken from hemispherical photographs.

The authors state: "While long-term microclimatic monitoring and consistent observational methods were employed to enhance the reliability of the data." Do the authors really believe that one year of monitoring constitutes "long-term" monitoring?

Also, "Our study uses a consistent, high-precision sampling methodology" What do the authors refer to "high-precision"?

Reviewer #3: Dear Authors,

I have carefully reviewed your responses to the reviewers' comments and the revised manuscript. I find that you have addressed all concerns raised by the reviewers thoroughly and appropriately. The revisions have improved the clarity and scientific soundness of the paper.

I am satisfied with your responses and the quality of the revised version.

I therefore recommend acceptance of your manuscript in its current form.

**Do you want your identity to be public for this peer review?** For information about this choice, including consent withdrawal, please see our Privacy Policy

Reviewer #1: No

Reviewer #3: No

---

## [Author Response · Author response to Decision Letter 3]

13 Jan 2026

Dear Editors and reviewer 1:

We sincerely thank Reviewer #1 for the careful evaluation of our manuscript and for providing valuable comments, which have greatly helped us improve the clarity, robustness, and broader relevance of our study. We have carefully considered each comment and revised the manuscript accordingly. We have primarily addressed the reviewer’s key concerns regarding:

1. Provide a clearer mechanistic explanation for why cooling differs among forest types (beyond reporting magnitudes). Add key canopy structural measurements (such as canopy cover/closure/openness from hemispherical photographs) to support the interpretation.

2. Strengthen the practical implications in the Introduction and Discussion, linking edge-to-interior thermal dynamics to forest management and species adaptation, and add specific ecological examples (e.g., understory biodiversity).

3. Add details on the snapshot replication timing and temporal coverage (dates and sampling duration).

4. Clarify and adopt more cautious terminology, such as “long-term” and “high-precision”.

Comment 1:

Strengthened mechanistic interpretation of the observed cooling differences among forest types by incorporating an expanded dataset, including newly collected understory temperature measurements with additional transect replication, stand/structural variables, and canopy structural metrics derived from hemispherical photographs (canopy openness, Openness, and leaf area index, LAI).

Response:

We agree with the reviewer that reporting cooling magnitudes alone is not sufficient without mechanistic interpretation. To address this, we expanded our dataset by adding forest structural metrics, including canopy openness (Openness) and leaf area index (LAI) derived from hemispherical photographs, together with additional stand/structural variables. We also incorporated short-term replicated transect temperature measurements to strengthen spatial support for the observed edge-to-interior patterns. Based on these additions, we provide an initial, structure-informed explanation for why cooling intensity differs among forest types. Corresponding revisions have been made in the Materials and Methods, Results, and Discussion, and we have updated the Abstract and Conclusions to reflect the enhanced mechanistic interpretation. We also explicitly acknowledge the remaining limitations and outline how future work (e.g., broader replication and concurrent process measurements) can further improve mechanistic attribution.

Materials and Methods

To assess whether the year-long automated transect captured spatially representative edge-to-interior patterns within each forest, we established four additional transects parallel to the primary transect for snapshot measurements. This resulted in five transects per forest when including the original transect, and the snapshot design is described in the Supplementary Material.

Spatial representativeness of the automated transect was evaluated using snapshot transects. The log-distance mixed model was refitted to the combined dataset that included both the year-long automated transect and snapshot measurements. Predicted edge-to-interior curves overlapped and remained within the 95% confidence intervals, indicating that the automated transect captured a spatially consistent gradient within each site (S4 Fig). Snapshot transects were collected in a different year and within a limited measurement window, so they were used only to assess spatial robustness and were not used to infer temporal dynamics.

Results

Spatial robustness was supported by snapshot transects. Snapshot derived offsets aligned with the automated transect predictions and remained within the 95% confidence bands, indicating that the automated transect captured a consistent edge to interior pattern within each site (S4 Fig). Supplementary structure analyses suggested that the distance response can be modulated by canopy structure in a site-specific manner, with significant negative interactions between log distance and LAI in the temperate and subtropical sites and between log distance and canopy openness in the temperate site (S1 Table).

Discussion

Supplementary structure analyses provide additional insight into how the edge gradient in maximum cooling can be shaped locally. Distance interactions with LAI were negative in the temperate coniferous and subtropical sites, and a negative distance interaction with canopy openness was detected in the temperate coniferous site (S1 Table). These results suggest that spatial changes in canopy density and openness along the transect can modulate how rapidly cooling strengthens from the edge to the interior [16,36,43,54]. Denser canopies can reduce radiative inputs and damp air exchange with the surrounding matrix, which promotes faster accumulation of cooling with distance [43]. More open edges can increase shortwave penetration and turbulent mixing, which weakens cooling near the boundary and delays the emergence of interior like conditions [16,29,55]. Although these structural signals were forest specific, they align with established mechanisms linking canopy structure and mixing to microclimate buffering [36,55,56].

Comment 2:

Strengthen the practical implications in the Introduction and Discussion, linking edge-to-interior thermal dynamics to forest management and species adaptation, and add specific ecological examples (e.g., understory biodiversity).

Response:

Thank you for this helpful comment. We agree that the practical relevance of edge-to-interior thermal dynamics should be more explicitly highlighted. Accordingly, we have reorganized and revised both the Introduction and Discussion to strengthen the management and species-adaptation implications of our findings, with particular emphasis on potential consequences for understory biodiversity. In addition, we added a dedicated subsection in the Discussion (Section 4.3) that synthesizes how DEI-based delineation of thermally decoupled interiors can inform the identification of microrefugia and support edge-aware management decisions.

Discussion Section 4.3 DEI-defined decoupled interiors as microrefugia and management implications

DEI-based delineation of coupled and decoupled zones provides a practical way to define microrefugia using thermal decoupling rather than a fixed distance alone. Across sites, decoupled interiors consistently exhibited stronger extreme cooling than coupled edge zones, which supports their functional interpretation as daytime thermal refugia under high heat load [58,59]. Cooling magnitudes in the decoupled interior were substantial and reached approximately 8°C in the tropical site. Such differences are large enough to shift thermal safety margins for temperature-sensitive understory organisms and to alter the conditions experienced during short-lived heat events [68,69].

Microrefugia defined in this way are inherently dynamic. Both the extent and, in some months, the presence of decoupled interiors varied strongly through the year. In the tropical site, a decoupled interior was not detected within 99.5 m during April and May. This pattern is consistent with a seasonal contraction of thermally decoupled interiors or with a DEI that exceeded the sampled transect [29,57]. Either interpretation implies reduced refuge availability during a period that can coincide with high thermal stress in the pre-monsoon season. Seasonal variability in canopy energy exchange offers a coherent explanation. Shifts in cloudiness, evaporative demand, and water availability can change the temperature contrast between open and forest environments and can modify how rapidly interior-like conditions emerge with distance from the edge [12,44,70].

This decoupling framework has direct management relevance because it translates microrefugia into a measurable spatial threshold. A forest patch can only provide a DEI-defined refugium if it contains area where the distance to the nearest non-forest edge exceeds DEI. When the maximum distance to edge within a patch is smaller than the relevant DEI, a thermally decoupled interior is unlikely to form, and the patch may fail to sustain the strongest cooling function even if mean offsets remain negative. This provides a clear basis for evaluating fragmentation impacts and for prioritising patch retention [35,71]. Distance-to-edge maps can be combined with DEI estimates to quantify the fraction of each patch that qualifies as potential refugial core under specific months or seasons [72].

Conservative design for year-round buffering can be based on the largest DEI observed across seasons or across critical months, whereas climate-adaptive planning can target DEI values for the periods when heat stress risk is highest. This time-specific approach avoids relying on a single annual DEI that may obscure months when decoupled interiors contract [51,52]. Because each forest type was represented by one focal site, the numerical thresholds reported here should be interpreted as evidence from a climatic gradient rather than universal limits. Replication across multiple sites per forest type and concurrent measurements of radiation, wind, and moisture would further strengthen mechanistic attribution and improve transferability of DEI-based guidance for edge-aware conservation planning in fragmented landscapes [11,13,73].

Comment 3:

Add details on the snapshot replication timing and temporal coverage (dates and sampling duration).

Response:

Thank you for the reviewer’s comment. We have now added full details on the snapshot replication in the manuscript and Supplementary. The community-structure variables and understory air temperature data were collected from 1–25 April 2025. Each forest site was surveyed for at least one week during this period. Measurements were conducted between 09:30 and 18:00, and we visited all plots at least twice within this window, with each measurement lasting 20 minutes.

Comment 4:

Clarify and adopt more cautious terminology, such as “long-term” and “high-precision”.

Response:

Thank you for the careful and constructive comment. We agree that the wording should be more precise. In the revised manuscript, we have replaced “long-term” with a more accurate description of the monitoring duration (i.e., one year of continuous monitoring) and we no longer refer to it as “long-term.” We also clarified that “high-precision” refers to the high temporal resolution and standardized deployment of the temperature loggers (e.g., consistent sensor type, placement height, radiation shielding, and logging interval) rather than implying unusually low measurement error. In addition, we have reorganized and partially rewritten the manuscript to improve terminology consistency and strengthen the logical coherence across sections.

Also, the response letter has been included in the uploaded files.

---

## [Decision Letter · Decision Letter 3]

15 Jan 2026

Dear Dr. Wang,

We look forward to receiving your revised manuscript.

Kind regards,

Lingye Yao, Ph.D.

Academic Editor

PLOS One

Journal Requirements:

Reviewers' comments:

Reviewer's Responses to Questions

**Comments to the Author**

Reviewer #1: All comments have been addressed

2. Is the manuscript technically sound, and do the data support the conclusions?

Reviewer #1: Yes

3. Has the statistical analysis been performed appropriately and rigorously?

Reviewer #1: Yes

4. Have the authors made all data underlying the findings in their manuscript fully available?

Reviewer #1: Yes

5. Is the manuscript presented in an intelligible fashion and written in standard English?

Reviewer #1: Yes

Reviewer #1: The authors have responded to the comments in detail.

These are my final minor comments before acceptance.

Minor comments

Line 97: These are not questions. Please revise for clarity.

Lines 160–161: The authors refer readers to the supplementary for details on the snapshots. However, they should also mention that forest structure measurements were conducted and are detailed in the supplementary (as some of those results are referenced in the discussion).

L270-272: This is not a result, and repeats the same statement from lines 211-213.

L371-373: Repeats the same statement of results from lines 326-329.

L389-391: Repeats the same statement of results from lines 272-275.

L411-413: Repeats the same statement of results from lines 295, 435.

Etc.

I find it problematic that the authors most of the time repeat the same phrases presented in the results in the discussion. The discussion section should not repeat the results section because its primary purpose is to interpret, synthesize, and contextualize the findings within broader scientific implications, rather than redundantly repeat the results numbers. Please review this in the discussion section.

**Do you want your identity to be public for this peer review?** For information about this choice, including consent withdrawal, please see our Privacy Policy

Reviewer #1: No

---

## [Author Response · Author response to Decision Letter 4]

16 Jan 2026

Dear Editors and reviewer 1:

We sincerely thank Reviewer #1 for the careful evaluation of our manuscript and for providing valuable comments. We have revised the manuscript accordingly and updated the line numbering throughout the tracked-changes version so that every edit can be located unambiguously. We have primarily addressed the reviewer’s key concerns regarding:

1. Clarifying the relevant wording and explicitly noting in the main text that forest-structure measurements were conducted for the snapshot transects.

2. Streamlining the Discussion to avoid repeating Results.

Detailed point-by-point responses are provided below.

Comment 1:

Line 97: These are not questions. Please revise for clarity.

Response:

Thank you for the careful comment. In the revised manuscript, we revised the relevant text by framing it explicitly as study objectives and introducing the three aims with “We address three aims:”, thereby improving clarity and consistency.

Comment 2:

Lines 160–161: The authors refer readers to the supplementary for details on the snapshots. However, they should also mention that forest structure measurements were conducted and are detailed in the supplementary (as some of those results are referenced in the discussion).

Response:

Thank you for this valuable comment. In the revised manuscript, we have added a description of the stand-structure measurements and hemispherical fisheye photography, and we clarify that the methods are provided in the Supplementary Material.

Materials and methods, Lines 160-164:

This resulted in five transects per forest when including the original transect. In addition, we conducted stand-structure measurements and took hemispherical fisheye photographs in all plots across the five transects to derive forest structural metrics, such as canopy openness (Openness) and leaf area index (LAI), and the snapshot design and methodological details are described in the Supplementary Material.

Comment 3:

L270-272: This is not a result, and repeats the same statement from lines 211-213.

Response:

Thank you for this valuable comment. We revised and de-duplicated the relevant text in the revised manuscript. In the Materials and Methods (Lines 214–215 in the revised manuscript; formerly Lines 211–213), we now describe only the snapshot-based procedure and criteria used to assess spatial representativeness, and we removed result-oriented statements. In the Results (Lines 272–273 in the revised manuscript; formerly Lines 270–272), we rewrote the text as objective result statements, avoided repeating the methods, and retained/clarified the additional findings from the supplementary structure analyses.

Materials and Methods, lines 214–215:

This fit was used to generate predicted edge-to-interior response curves and corresponding 95% confidence intervals for each site (S4 Fig).

Results, lines 272–273:

Snapshot derived offsets aligned with the automated transect predictions and remained within the 95% confidence bands across sites (S4 Fig).

Comment 4:

L371-373: Repeats the same statement of results from lines 326-329.

Response:

Thank you for this comment. In the revised manuscript, we removed the result-repeating numerical statements (including the peak-month listing) from the Discussion and retained only a brief qualitative statement.

Discussion, lines 371-372:

Seasonal timing also mattered, and the strongest decoupled interior cooling occurred in different months across sites (Table 2).

Comment 5:

L389-391: Repeats the same statement of results from lines 272-275.

Response:

Thank you for this comment. In the revised manuscript, we removed the repeated interaction statements from the Discussion and retained only a brief qualitative statement.

Discussion, lines 386-386:

Supplementary structure analyses suggest that local variation in canopy structure may influence how rapidly maximum cooling strengthens from the edge to the interior (S1 Table).

Comment 6:

L411-413: Repeats the same statement of results from lines 295, 435.

Response:

Thank you for the careful and constructive comment. In the revised manuscript, we removed this repeated result statement from the scale-dependence paragraph (Lines 411–413) and retained it only once where it is interpreted mechanistically (Lines 435 in the revised manuscript; formerly Lines 428-432).

Discussion, lines 428-432

In contrast, in TF during April–May, edge–interior coupling appears to be especially strong, such that DEI may extend beyond the sampled 99.5 m transect. This pattern is consistent with reduced canopy cooling capacity and/or enhanced coupling to the surrounding matrix [11,68,69], which could shift DEI beyond the sampled extent.

In addition to the revisions made in response to the reviewer comments, we also further streamlined the Discussion to improve clarity and avoid redundancy. Specifically, we revised the text formerly in Lines 452–454 (now Lines 447-449 in the revised manuscript) and removed overlapping wording that repeated statements already presented elsewhere.

Discussion, lines 447-449:

Microrefugia defined in this way are inherently dynamic. Both the extent and, in some months, even the presence of thermally decoupled interiors varied markedly through the year, indicating that refugial availability is time dependent.

In addition, the response letter has been included in the uploaded files.

---

## [Editor Report · Decision Letter 4]

19 Jan 2026

Spatiotemporal variation of the maximum cooling effect across edge-to-interior gradients in forest patches of southwestern China

PONE-D-25-08797R4

Dear Dr. Wang,

We’re pleased to inform you that your manuscript has been judged scientifically suitable for publication and will be formally accepted for publication once it meets all outstanding technical requirements.

Kind regards,

Lingye Yao, Ph.D.

Academic Editor

PLOS One
---

## [Editor Report · Acceptance letter]

PONE-D-25-08797R4

PLOS One

Dear Dr. Wang,

I'm pleased to inform you that your manuscript has been deemed suitable for publication in PLOS One. Congratulations! Your manuscript is now being handed over to our production team.

Kind regards,

on behalf of

Dr. Lingye Yao

Academic Editor

PLOS One